# Community health workers experiences and perceptions of working during the COVID-19 pandemic in Lagos, Nigeria—A qualitative study

Zahra Olateju[1]*, Tolulope Olufunlayo[2], Christine MacArthur[3], Charlotte Leung[1], Beck Taylor[4]

1 College of Medical and Dental Sciences, University of Birmingham, Birmingham, England, United Kingdom, 2 Department of Community Health and Primary Care, College of Medicine, University of Lagos, Yaba, Lagos State, Nigeria, 3 Department of Maternal Health, Institute of Applied Health Research, University of Birmingham, Birmingham, England, United Kingdom, 4 Department of Public Health, Institute of Applied Health Research, University of Birmingham, Birmingham, England, United Kingdom

☯ These authors contributed equally to this work.
* zxo775@student.bham.ac.uk

**Data Availability Statement:** The data that support the findings of this study are not publicly available due to them containing information that could compromise research participant privacy. The

## Abstract

### Background

Community Health Workers are globally recognised as crucial members of healthcare systems in low and middle-income countries, but their role and experience during COVID-19 is not well-understood. This study aimed to explore factors that influence CHWs' ability and willingness to work in the COVID-19 pandemic in Lagos.

### Design

A generic qualitative study exploring Community Health Workers experiences and perceptions of working during the COVID-19 pandemic in Lagos, Nigeria.

### Methods

15 semi-structured, in-depth, video interviews were conducted with Community Health Workers purposively sampled across seven of Lagos' Local Government Areas with the highest COVID-19 burden. Interviews explored Community Health Workers' attitudes towards COVID-19, its management, and their experiences working in Lagos. Data was analysed thematically using the framework method.

### Results

Three main themes were identified. *1. Influences on ability to undertake COVID-19 Role*: Trust and COVID-19 knowledge were found to aid Community Health Workers in their work. However, challenges included exhaustion due to an increased workload, public misconceptions about COVID-19, stigmatisation of COVID-19 patients, delayed access to care and lack of transportation. *2. Influences on willingness to work in COVID-19 Role*: Community

authors did not seek ethical permission from the participants, nor the ethics committee, for the data to be used for anything other than this particular research study. The authors therefore do not have explicit permission for data sharing, re-analysis nor future studies and so would be inappropriate and unethical to make them available in the public domain. While anonymised, the data contains potentially identifying patient information. However, qualified individuals can direct queries by contacting Dr Ruth Riley (r.riley@bham.ac.uk) - chair of the University of Birmingham BMedSci Intercalation Internal Ethics Review Committee.

**Funding:** The author(s) received no specific funding for this work.

**Competing interests:** The authors have declared that no competing interests exist.

Health Workers' perceptions of COVID-19, attitudes towards responsibility for COVID-19 risk at work, commitment and faith appeared to increase willingness to work. *3. Suggested Improvements*: Financial incentives, provision of adequate personal protective equipment, transportation, and increasing staff numbers were seen as potential strategies to address many of the challenges faced.

## Conclusion

Despite Community Health Workers being committed to their role, they have faced many challenges during the COVID-19 pandemic in Nigeria. Changes to their working environment may make their role during disease outbreaks more fulfilling and sustainable. International input is required to enhance Nigeria's policies and infrastructure to better support Community Health Workers during both current and future outbreaks.

## Introduction

On 30th January 2020, Coronavirus Disease-19 (COVID-19) was declared an international emergency [1]. By February 2020, the virus had spread to the Sub-Saharan country Nigeria which [2], as of August 10, 2021, had reported over 178,000 cases and 2,187 deaths [3].

Access to healthcare in Nigeria remains a profound problem [4]. Socioeconomic inequalities and geographical inaccessibility across both urban and rural communities have caused unequal healthcare use across the country [4]. Community Health Workers (CHWs) play a key role in trying to combat this problem. As trained members of the public, they use their skills to provide healthcare, and to educate their communities [5,6]. There are different types of CHWs, each receiving varying levels of training. Community health extension workers (CHEWs) receive two to three years of "health-related training", approved by the Community Health Practitioners Registration Board of Nigeria [7]. This includes over 90 hours of lectures and "competency-based training" which gives them the skills to provide basic primary care as recognised by Nigeria's national policy [7]. Other CHW types receive training focused on improving primary health care delivery in Nigeria [8]. This includes training on: "environmental sanitation", "immunisation", "health education and community mobilisation", "treatment of minor ailments", "water and sanitation", "nutrition and growth monitoring" and "essential drug supply" [8]. CHWs' contribution to healthcare, particularly within low and middle-income countries (LMICs), has been globally recognised [9–12]. They provide care such as antenatal services, HIV and TB management, maternal, neonatal and child services and routine immunisation [9,13]. Their close relationships with community members, who are often underrepresented in society [11,12], aid in bridging the gap between the community and the healthcare systems [14]. By raising community awareness and implementing contact tracing, CHWs have proven to be imperative in limiting viral transmission during pandemics [9,10,15,16]. Throughout the 2014–6 Ebola outbreak, they acted as contact tracers and active case finders, whereby any suspected cases found were isolated and referred to treatment centres [17]. More recently, during the COVID-19 pandemic, CHWs have aided in the home delivery of medications [18], follow up of patients [19], health education [20], nutrition screening, vaccine delivery [21] and psychological and social support [22,23].

Despite Lagos reporting a high number of COVID-19 cases, there are fears that these figures may underrepresent the true impact of the pandemic [24]. It is believed that due to the

social and economic challenges of COVID-19, many undetected cases exist within communities [24,25]. The longer-term effects of this are becoming a growing concern due to its potential adverse health effects [26,27]. Provision of healthcare services for those with non-COVID-19 related health problems, such as malaria, tuberculosis and cholera, has also reduced substantially since the start of the pandemic [26,27]. COVID-19 pressures mean that many struggle to access care due to lockdown measures, restricted movement and reduced household income resulting from limited work opportunities [26]. Additionally, as uptake of healthcare has decreased due to public fear of infection, increased morbidity and mortality from a spectrum of diseases is expected [26]. Whilst the ultimate consequence of this is yet unknown, the potential effects could be damaging, and understanding and supporting the role of CHWs in health systems is arguably more important now, than it has ever been.

In 2020, Mayfield-Johnson et al conducted a focus group study with seven CHW leaders in the United States to explore the impact of COVID-19 on their role, which suggested that CHWs' fear of contracting COVID-19 reduced their motivation, though they remained resilient to continue to support community members [11]. However, as CHWs are integral in healthcare delivery in LMICs such as Nigeria, research needs to be conducted in these settings to understand the effects of the COVID-19 pandemic on their role [9]. Earlier in the pandemic, a rapid evidence synthesis had suggested that incentives such as bonuses may be useful in increasing CHWs' motivation [9]. More recent expert opinion suggested that the main challenges faced by CHWs in Nigeria were "mistrust of political entities", overwhelming workload, "stigma and misinformation", "limited testing capacity" and "poor adherence to quarantine and isolation" [28]. However, we have not identified any published qualitative interview studies with frontline CHWs in Nigeria which explore their perspectives and experiences.

## Materials and methods

### Aims and objectives

This study aimed to explore factors that influence CHWs' ability and willingness to work in the COVID-19 pandemic in Lagos, Nigeria, and to identify how to better support CHWs during disease outbreaks.

### Study design

This was a generic qualitative study [29] that involved conducting remote semi-structured interviews to elicit an in-depth understanding of participants' views [30]. A generic study design was chosen, guided by our research aims and provide a rich description of the perspectives of CHWs regarding their willingness and ability to work, and support needs throughout the pandemic, without claiming an "explicit or established set of philosophic assumptions" [29,31].

### Setting

This study was conducted in Lagos, Nigeria's most populous city, with over 21 million citizens from many ethnic backgrounds [32]. It is located in the south-west of the country and is divided into five administrative divisions comprising of 20 Local Government Areas (LGAs), which can be categorised as either urban or rural [33]. As of June 9, 2021, Lagos reported over 59,000 COVID-19 cases [34]. LGAs contributing a high proportion to the COVID-19 burden in the state were included in the study (Lagos Mainland, Surulere, Ikeja, Alimosho, Eti-Osa, Kosofe and Mushin) [35,36].

## Sampling

CHWs were sampled purposively from LGAs with a high COVID burden, and further participants were recruited using snowball sampling to ascertain data-rich interviews [37–39]. To allow the Principal Researcher (ZO) to interview until data saturation, the research team aimed to recruit 15–20 participants hence both sampling techniques were used [40]. All cadres of CHWs were sought who fitted the eligibility criteria which were: over 18 years, able to speak fluent English and a CHW who worked during the COVID-19 pandemic in Lagos.

## Recruitment

Recruitment and data collection occurred simultaneously during March and April 2021. Due to the COVID-19 pandemic, recruitment was conducted remotely with the aid of a senior researcher based in Lagos. The research team recruited participants through the Medical Officers of Health, who helped to identify CHWs who fitted the eligibility criteria. Once identified they helped to establish appropriate communication channels for recruitment. In addition, the senior researcher contacted CHWs whom she knew from previous research projects. To ensure a diverse representation, both salaried and voluntary CHWs were recruited across both sexes. Interested individuals were contacted, screened for eligibility, sent a copy of the participant information leaflet (PIL) and consent form prior to their interview before giving informed consent. Researchers completed the PLOS inclusivity in global research questionnaire, as this research was conducted outside the researcher's own country (S2 Fig).

## Data collection

Semi-structured interviews were conducted following a topic guide (S1 Fig) created from research team input. Topics included personal and societal attitudes to COVID-19, experiences of working during the pandemic and management of the pandemic in Lagos. A pilot interview, with a senior researcher (TO), was conducted for topic guide refinement [41]. Data from this was not included in analysis. Remote interviews, using Zoom, were conducted by ZO, based in the UK, in English. Interviews were video- and audio-recorded. After each interview, participants were compensated with a phone card voucher and ZO documented their initial impressions which aided in the identification of the point of data saturation as well as identifying developing themes [40].

## Data analysis

Data analysis was inductive and iterative which allowed for topic guide refinement for subsequent interviews. Thematic analysis was undertaken using the framework method [42]. This consisted of seven stages of analysis: Transcription; familiarization with the interview; coding; developing a working analytical framework; applying the analytical framework; charting data into the framework matrix; and interpreting the data. Initially, ZO transcribed all interviews verbatim. Following this, ZO became familiar with the data by listening to interview audio recordings, reading the transcripts at least twice and reading the field notes. Subsequently, three transcripts were inductively independently open coded by ZO and a senior researcher (BT), and a further two transcripts by ZO and another researcher (CL). Inconsistencies in coding were discussed and addressed. Following coding of initial transcripts, a working, hierarchical analytical framework was developed in collaboration with the wider team, and ZO applied this framework to the rest of the dataset. Subsequently, data was charted into a framework matrix, incorporating the data from the demographic questionnaire to contextualise findings. Following this, ZO used the framework matrix to interpret the data and develop themes,

working with the wider team to reflect and refine interpretation. Data management and analysis was facilitated using QSR NVivo 12.

The research team adopted a reflexive, inductive process to ensure transparency in the study methods, consider how they may have influenced the data collection and analysis processes, and reduce bias [43,44].

## Ethics statement

Ethical Approval for this study was granted by the University of Birmingham Internal Ethics Review Committee (Reference: IREC2020/1760475) and Lagos University Teaching Hospital Health Research Ethics Committee (Reference: ADM/DCST/HREC/APP/4054). Permission to work with CHWs was granted by the Lagos State Primary Health Care Board. Before the interviews, verbal consent was obtained from all participants after explaining the research to them.

## Results

There were 15 participants interviewed, with interviews lasting on average 46 minutes. They included four Community Health Officers (CHOs), six CHEWs, four Community Mobilisers, and one Community Volunteer, with an average age of 42 years. Participant roles before the pandemic included door-to-door health education and delivering healthcare at community clinics. However, during the pandemic, most participants took on additional responsibilities. In general, CHEWs, Community Mobilisers and Volunteers were required to educate community members on COVID-19, undertake active case search, contact tracing, refer individuals for a COVID-19 test, deliver medications and administer the COVID-19 vaccination. CHOs had a managerial role supervising other CHWs and ensuring that COVID-19 patients received at-home care. The distribution of participant demographics is shown below in *Table 1*.

## Findings

Three themes were developed after data analysis: (1) 'Influences on ability to undertake COVID-19 Role' and (2) 'Influences on willingness to work in COVID-19 Role' and (3) 'Suggested Improvements'. Each theme had several sub-themes as shown in Table 2.

**Table 1. Participant demographics.**

| Patient Demographics | Characteristics | N = 15 |
|---|---|---|
| Gender | Female | 12 |
| | Male | 3 |
| Age range (years) | 20–30 | 1 |
| | 31–40 | 5 |
| | 41–50 | 5 |
| | ≥51 | 4 |
| Highest Level of Education | Senior secondary education | 1 |
| | Tertiary education | 14 |
| Occupation | Community Health Extension Worker | 6 |
| | Community Health Officer | 4 |
| | Community Mobiliser | 4 |
| | Community Volunteer | 1 |

**Table 2. Summary of themes and subthemes.**

| THEME | SUBTHEMES | CODES | FREQUENCY OF CODES BY PARTICIPANTS |
|---|---|---|---|
| 1. Influences on ability to undertake COVID-19 Role | a) Trust<br>b) CHW Knowledge about COVID-19<br>c) Exhaustion<br>d) Stigma<br>e) Public misconceptions about COVID-19<br>f) Delayed Access to Care<br>g) Lack of Transportation | Trust with Community<br>Participant COVID Knowledge<br>Draining<br>Stigma<br>Misconceptions<br>Fears of being diagnosed<br>Fears of getting infected<br>Transportation | 8<br>15<br>11<br>4<br>10<br>7<br>3<br>1 |
| 2. Influences on willingness to work in COVID-19 Role | a) CHW perceptions of COVID-19<br>b) CHW perceptions of Government COVID-19 Response<br>c) CHW attitude to responsibility for COVID-19 risk at work<br>d) Commitment to Their Role<br>e) Faith | Disruptive<br>Low risk<br>Threat<br>Terrible<br>Adaptation<br>Government<br>Uncertainty<br>Positive outlook<br>Personal responsibility<br>Sense of duty and responsibility<br>Enjoyment<br>Faith | 5<br>11<br>8<br>10<br>4<br>146210141112 |
| 3. Suggested Improvements | a) Financial Incentives<br>b) PPE<br>c) Transportation<br>d) More Staff | Financial<br>PPE<br>Transportation<br>More Staff | 11<br>10<br>4<br>1 |

## 1. Influences on ability to undertake COVID-19 Role

**Trust.** Several participants felt that their pre-existing trust with community members significantly aided them in asking questions, obtaining information, and educating communities about COVID-19 and its prevention. As members of the communities they were educating, participants could often relate to the population, speak their language, and successfully immerse themselves within the community.

*"People in my community know me so well that is why it was a bit easy for me to ask them questions"* [P1]

**CHW knowledge about COVID-19.** All participants demonstrated broad knowledge of COVID-19 due to the training they received. However, one participant believed that the virus affects the liver. Most participants were aware of the virus' mode of transmission, with sneezing, direct contact with an infected person and cough being frequently referenced. The elderly, frontline workers, and individuals with comorbidities were seen as more at-risk due to a diminished immune response and increased viral exposure. This knowledge aided them in delivering health education as well as ensuring that at-risk individuals were quickly tested for COVID-19 and received appropriate healthcare.

*"COVID-19 is a very deadly disease and it is transmitted through [. . .] sneezing and [. . .] maybe the person did not wash their hands and they use their hand to touch any surface, anybody that touch the surface contact it, so we enlighten our people about all this"* [P12]

**Exhaustion.**   Participants found their role during the pandemic physically and emotionally draining, due to the extra responsibilities COVID-19 added to their normal role. Participants also described the negative personal impact of patients not listening to CHW advice, and community members expecting provision of PPE from CHWs or the Government.

*"Oh my experience [. . .] it wasn't really fun I tell you it [. . .] wasn't funny it was [. . .] draining physically draining emotionally"* [P5]

**Stigma.**   A few participants discussed social stigmatisation of COVID-19 patients which caused many individuals not to disclose their COVID-19 infection when presenting to facilities, increasing participants' potential exposure. Consequently, CHWs reported difficulties in identifying COVID-19 cases in the community. The psychological impact of stigmatisation was highlighted by one participant who discussed that COVID-19 is viewed as a *"death sentence"* relating how some patients reported depressive symptoms following diagnosis. Therefore, continuous patient follow-up was seen as essential.

*"Even those that confirmed, they do not want to come out and say they have it because of fear of stigmatization too"* [P5]

*"We have to follow them or some of them once they get home [. . .] say I have [. . .] depression, depression might set in because they go home and be thinking that this is the end of their life"* [P8]

**Public misconceptions about COVID-19.**   Tackling public COVID-19 perceptions was a major challenge emphasised by all participants. Some individuals were reported not to believe that COVID-19 was a reality, making it harder for CHWs to encourage adherence to Government guidelines, with some participants reporting arguments with people who refused to follow guidelines. Participants attributed this to individuals believing it only exists in other countries, limited exposure to COVID-19 patients, and confusing COVID-19 with malaria due to similar disease presentations.

*"They believe that there's nothing like COVID even some people will tell you it's just malaria it's severe malaria"* [P13]

However, the most referenced factor was mistrust in the Government. Many participants expressed that some individuals believed that COVID-19 was being used by the Government to *"scam"* them.

*"That the Government are using it to embezzle money some that say there is nothing like COVID-19 in Nigeria generally it is existing abroad"* [P9]

Participants stated that, in the previous few months, public misconceptions surrounding COVID-19 vaccination had become more prevalent. Misconceptions included vaccine risks, such as blood clots, post-vaccination breakthrough infection, and thinking that preventative measures do not need to be followed by vaccinated individuals. Therefore, participants found increasing public COVID-19 vaccination uptake difficult. Social media was seen as the main source of public misinformation, hence several participants stressed the need for CHWs to continuously educate communities with the right COVID-19 information to minimise the impact of such beliefs, adding further responsibility to the CHW role.

"*After sensitizing a particular community just give them few days. By the time you [. . .] go back to see [. . .] how far they've been able to make use of what I've told them you'll discover that you have to start all over again*" [P5]

Some participants emphasised that the public frequently did not adhere to COVID-19 protocols due to hunger within communities, people experiencing breathing difficulties when wearing face masks, and unaffordable PPE. These factors often made it difficult for CHWs to encourage individuals to follow disease guidance.

"*People have [. . .] to eat, their children have to be taken care of, their home has to be taken care of, so it wasn't that easy for them to [. . .] really comply with all the guidelines*" [P4]

**Delayed access to care.**   Several participants highlighted that in the initial months of COVID-19 many patients were not accessing healthcare facilities due to fears of being diagnosed, getting infected, and/or not being attended to by healthcare professionals. Participants expressed that this made some patients unreachable, requiring more frequent CHW visits to the community. Patients were also reported to frequently present late with more severe symptoms, making it difficult for CHWs to perform in their role. However, participants acknowledged that through their community sensitisation and health education, access to care has significantly increased.

"*During the COVID period [. . .] the turnout was very poor because they believe that anybody that came to Government hospital during the COVID period we are going to diagnose them with COVID*" [P13]

**Lack of transportation.**   Three participants working in rural LGAs expressed that, during the initial lockdown, finding transportation to their duty post was extremely difficult, especially without access to their own vehicle. This was concerning for participants, as they could not perform in their role without access to the community.

"*But others that don't have vehicle [. . .] it is very difficult for them to get to [. . .] their place of work*" [P13]

## 2. Influences on willingness to work in COVID-19 Role

**CHW perceptions of COVID-19.**   All CHWs' views towards the pandemic were dependant on personal experience. Several participants acknowledged it disrupted people's livelihoods, with the most discussed impacts being redundancies, closed businesses, families struggling both financially and socially, and inflation of commodity prices. Despite participants' initial overarching negativity, they expressed a temporal change in their COVID-19 risk perception. Current COVID-19 prevalence in Lagos was perceived to be low, and participants were less concerned about COVID-19 compared to February 2020. Therefore, CHWs were more comfortable and eager to work, as they were less worried about their own personal chance of becoming infected.

"*Interviewer: And how would you feel if [. . .] you were diagnosed with COVID [. . .]?*

*[P1]: Before now if you had asked me this I would've said oh my God I'm gonna die but now no*"

Several participants highlighted that this temporal change was linked to a few underlying influences, including participants comparing COVID-19 to previous epidemics such as Ebola

and Yellow Fever. Participants' perceptions varied from some considering COVID-19 to be the worst epidemic, and others thinking to the contrary. Other influences discussed included COVID-19's novelty, and the limitations of the Nigerian healthcare system, which increased participants' perceived threat of COVID-19. CHWs who knew individuals previously infected with and recovered from COVID-19 often viewed it as less of a threat, as they felt that, should they get infected, it would be non-life threatening, furthering their eagerness to work.

> "*Ebola one is harder this COVID-19 is not all that hard if you can just obey what they have told you to do the social distance, personal hygiene, face mask [. . .] I feel COVID-19 is still better*" [P8]

> "*Somebody told me that he got infected but now he is fine so I won't feel any bad*" [P1]

An overarching theme across all participants was having adapted to life in COVID-19, which manifested in following COVID-19 protocols, changing hobbies, and general lifestyle activities. As participants had adjusted, they had become more willing to work, due to increased familiarity with the pandemic and its restrictions.

> "*The COVID-19 pandemics is [. . .] a global phenomenon that has changed our [. . .] lifestyle to a new normal we are presently adapting to a new lifestyle*" [P14]

**CHW perceptions of government COVID-19 response.** Many participants acknowledged that the Government had tried their best to tackle COVID-19, by setting up isolation centres and distributing food, and free home care kits to infected individuals. They also cited Government collaboration with organisations, such as the African Centres for Disease Control, the World Health Organisation and the Nigerian Centre for Disease Control. However, several participants highlighted that due to Nigeria's large population and limited resources, authorities could not support everyone. A sense that the authorities were trying their best appeared to encourage some participants to try their best in their role too.

> "*When it comes to Government [. . .] they can't help everybody [. . .] the population the resources are not enough to go around*" [P9]

Body language from several participants suggested that they were hesitant to criticise the Government's COVID-19 response. When asked, one participant responded with: "*I don't want to answer it.*" However, another participant discussed that more could be done to ensure the Government leads by example, as there were some occasions where authorities did not adhere to COVID-19 protocols. Whilst not explicitly stated, this could be important in encouraging public adherence to COVID-19 guidelines, and CHWs' willingness to work. In particular, they mentioned that during election campaigns some individuals were not complying with COVID-19 guidelines.

> "*An election where you allow a candidate to partake in campaigning without observing the protocol and election where you converge people [. . .] at your own expense, [. . .] because of the political gain and you're saying that social distance should be observed*" [P14]

**CHW perception of COVID-19 risk in work role.** The majority of participants demonstrated awareness of their increased risk of COVID-19 exposure through their role, and how there was uncertainty surrounding their exposure due to interacting with patients who have

not yet had a COVID-19 test. Despite this, participants were still willing to work as they felt that exposure to diseases was part of their role as healthcare workers.

> "*To the exposure [. . .] as a frontline health worker we can't say we are not exposed and we cannot say we are exposed*" [P2]

Some participants expressed fear due to COVID-19's unfamiliarity. However, many participants had previous experience working in disease outbreaks, so they remained willing to work. Some participants discussed being concerned about their own personal COVID-19 risk due to a lack of beds in Nigerian hospitals. These participants also remained willing to work, though they reported using more PPE.

For several participants, more focus was placed on taking personal responsibility for their infection control and COVID-19 risk. Employers provided PPE for most CHWs, but this was often deemed inadequate, leading to many participants buying their own protection and changing their health behaviours. This was reported particularly by participants with comorbidities, or at-risk family members at home. Some participants expressed that the inadequate provision of PPE by employers negatively impacted their desire to work, as they felt that their employers should protect them in their role. Most participants gave accounts which suggested that they were falsely reassured by wearing surgical face masks. Most reported that they were protecting themselves and were not aware that they were predominately protecting others.

> "*I have not been infected since it started because normally prevent myself [by] wearing mask, at least I double mask*" [P11]

**Commitment to their role.**   All participants expressed a sense of duty and responsibility which was a driving factor in furthering their willingness to work. Many participants felt that as they had been trained for their role, they were needed by their community.

> "*You need to be able to assist that's why you are trained as a health worker you must help others [. . .] I need to do my job*" [P15]

For a few participants, their *"passion"* for the role furthered their eagerness as they loved being a CHW. In particular, one participant discussed that they were willing to work despite the personal risk.

> "*Because um I love health. I love taking care of people. [. . .] I love people being well I don't like people you know I just love being in my community, [. . .] as a community health personnel*" [P4]

**Faith.**   Many participants also acknowledged their religion played a crucial role in reassurance whilst working. Participants believed that their fate was in God's hands, lowering their perceived threat of COVID-19, hence increasing their willingness to work.

> "*Just by his Grace [. . .] I ain't worried*" [P6]

## 3. Suggested improvements

**Financial incentives.**   Participants all stated that improvements are needed to make the role less strenuous and more rewarding. However, no participant said that they would

consider leaving the role if improvements were not made (though this was not asked explicitly). The most discussed improvement was financial incentives. Whilst not all participants were salaried CHWs, many felt a form of financial recognition was warranted due to the challenging nature of the role. Most participants stated that despite already receiving an *"added allowance"*, an additional COVID-19 payment given as recognition for their work, they felt it was not enough and *"not worth the risk"* they had been exposed to.

*"They should increase our added allowance. Our added allowance is very poor so [. . .] they should work on that"* [P2]

**PPE.** Insufficient PPE provided by employers was deemed unacceptable, due to the considerable COVID-19 risk in the CHW role, as previously mentioned. Therefore, provision of adequate PPE was suggested, to ensure CHWs are protected at all times.

*"They could've supported us better with PPE instead of us getting it ourselves"* [P1]

**Transportation.** Due to the lack of transportation previously discussed, some participants suggested that the provision of transportation for CHWs to and from their duty post during lockdowns, particularly in rural locations, would be invaluable.

*"They could have done to better support us is number one is they will have provide a form of ambulance to take you to your work place and return you back to your house"* [P9]

**More staff.** Additionally, one participant discussed that due to CHW redeployment to perform community searches, which involves searching for active COVID-19 cases, more staff were required to lighten the individual workload and aid in the continuation of care.

*"I told you we are short of staff a lot because [. . .] they are assigned to do Community search"* [P8]

## Discussion

Trust between participants and their community was reported to facilitate the CHW role. This is consistent with existing literature, which suggests CHWs' close relationships with community members helps to narrow the gap between the healthcare system and community [11,12,14]. In our study, the public were more open and welcoming to participants, aiding their ability to get 'inside information', quickly access active cases in the community, and swiftly perform contact tracing and successful health education. This is particularly important in LMICs, as the evidence suggests that many COVID-19 cases remain in the community [24]. Therefore, by working with the public, community transmission, future morbidity and mortality can be reduced [24–27].

Health education was a key component of participants' role aided by their broad COVID-19 knowledge gained from their pandemic training. Some knowledge inconsistencies were apparent, such as which system is affected by COVID-19 and the benefits of using face masks [45–47]. Formal knowledge testing in other studies has suggested knowledge gaps, which were not explored in depth in our study [48]. Future training opportunities should aim to rectify this to ensure the correct information is disseminated into the community. Existing quantitative literature has illustrated how CHWs are imperative in increasing public adherence to infection control measures through health education [16]. Our qualitative findings are

consistent with this, with CHWs reporting that health education improved public adherence to guidance by addressing misconceptions. Additionally, as many of the public deferred accessing care due to fears of infection, diagnosis, or not being attended to, health education proved vital in increasing health-seeking behaviours. This is consistent with an existing study however [27], contradicts other studies which suggest that the provision of healthcare services for non-COVID-19 related problems was scarce in Nigeria [25,26]. Participants expressed that healthcare was always available, however they found that many chose not to access. A key finding was that continuous health education appeared to be required to sustain adherence to guidance. Many participants expressed that re-educating communities is needed to address new misconceptions or misinformation penetrating communities.

The exhausting nature of the CHW role was a consequence of several other challenges faced including social stigmatisation and inaccurate public COVID-19 perceptions. This follows previous studies which suggested that not only are CHWs facing an overwhelming workload but also misconceptions and mistrust are rife within communities, requiring CHWs to remain resilient [11,28]. However, due to the sudden nature of disease outbreaks, such challenges can be expected. Social stigmatisation during pandemics is common [49], therefore it is important for authorities to disseminate the correct information to the public to reduce stigmatisation and disbelief. However, tackling public mistrust of authorities is imperative in improving future epidemic outcomes and CHW experience. It may be beneficial for future research to explore the origins of mistrust, so that strategies can be implemented to redevelop trust between the community and authorities.

Willingness to work was strongly influenced by CHWs' sense of duty, passion and faith, despite their increased exposure. A quantitative study suggested that fears of COVID-19 infection diminished CHW motivation for the role [50]. This study, however, has found that CHWs remained committed, and took personal responsibility for their own infection control, especially if their perceived COVID-19 risk was high, by ensuring they were equipped with PPE, despite it not always being provided [10]. This contradicts a previous study, which suggested CHWs do not protect themselves properly when exposed to COVID-19 patients [48]. However, it is consistent with other work which suggested that individuals with a higher perceived risk of infection engage more in preventative measures [51,52]. Future research exploring the use of PPE in CHWs is needed to ensure they are properly protected, with the correct PPE, especially during times when their perceived COVID-19 risk is low. Faith also appeared to play an important role in decreasing participants' worries about working, through prayer, which has not yet been explored by current studies.

Participants' COVID-19 perceptions were influenced by the novelty of COVID-19, comparing COVID-19 to previous epidemics, limitations of the Nigerian healthcare system, and knowing individuals who had recovered from a COVID-19 infection. Such factors, according to the author's searches, have not yet been explored by current literature, therefore provide novel insight into factors influencing CHWs' willingness to work. Time, however, had the largest influence. In the long term, COVID-19 was viewed as a low risk which may be due to increased familiarity and experience, leading to participants feeling more comfortable working. Similar associations between time and risk perception have been seen in previous studies, and positively correlate with increased healthy behaviours [53,54]. Awareness of these factors may help employers better support CHWs during pandemics. However, these factors are individual and country-specific. Most participants suggested they held positive perceptions towards the Nigerian Government COVID-19 response. This proved important in furthering their willingness to work and increased adherence to Government policies, which aligns with existing work [55]. However, participant hesitancy to comment on the Government's response

did not allow for full exploration, so future research exploring CHWs' true perceptions would be beneficial.

Financial incentives, sufficient PPE, transport provision, and increased numbers of CHWs were the main improvements suggested by participants. A previous study suggested that financial incentives increase motivation in CHWs; however [9], as this study evidenced, incentives need to be viewed as worthwhile by CHWs to have a noticeable impact on motivation. A recent study discussed the need for adequate PPE provision for frontline workers [10]. However, widespread PPE provision had not been implemented across Lagos at the time of writing. To the authors' knowledge, the other suggested improvements have not yet been explored by existing literature, therefore offers novel insight into bettering the CHW response.

Infrastructure changes are required to reduce poverty, hunger and make PPE more affordable, aiding in increasing adherence to disease policies [56,57]. Implementing rapid hiring, training, and deployment of CHWs in emergencies may help to improve their work environment, reduce individual workloads and manage public stigmatisation and misconceptions. Faster purchasing and distribution of PPE, transport provision for CHWs working in rural settings, and more training are also required to better prepare CHWs to face challenges previously experienced in epidemics [9,58]. However, long-term funding will be required as these suggestions may not be financially viable in Nigeria [59]. Policymakers should prioritise improvements based on need, and work with non-governmental organisations to fund and implement them. This could have wide-ranging benefits to the Nigerian health system, as a country with recurrent disease outbreaks, as well as better equipping CHWs to work in current and future epidemics.

## Strengths and limitations

This study has provided novel insight into CHWs' experiences and perceptions of working during COVID-19 in Lagos. Participant diversity was maximised by purposively recruiting male and female participants across urban and rural LGAs.

The majority of participants were female; however there did not seem to be any gender-based differences in findings. Participants were restricted to CHWs fluent in English, therefore findings may not be representative of all CHWs in Lagos. The involvement of CHWs in the COVID-19 vaccination program meant that fewer CHWs were available for recruitment, and it wasn't possible to recruit CHWs from all cadres. However, of the 17 individuals approached, 15 were available. As both purposive and snowball sampling were used, our results may not indicate the opinions of the majority of CHWs in Lagos. Understanding of cultural context in transcripts and developing themes were provided by TO, a senior Nigerian public health doctor. The study was conducted remotely and there were occasions where the internet connection was interrupted, and verbal communication was less clear. Power imbalances between participants and the principal researcher, a Black, female medical student, and the links with a senior public health professional and academic institution may have affected CHWs' willingness to speak freely. To mitigate this, rapport building was facilitated, participants were encouraged to speak openly, and reassured about confidentiality. In addition, field notes and regular debrief meetings facilitated adoption of a reflexive process. Data saturation was not fully reached as some new information was identified in the final interviews. Therefore, additional studies are needed to further explore the experiences and perceptions of CHWs which the current study was not able to cover.

## Conclusion

This study has highlighted that CHWs are a committed and crucial part of the Nigerian pandemic response. They remained dedicated and central to the success of its pandemic policies

through public health education, active case finding, and vaccinations among other commitments. More specifically, this study demonstrated the factors affecting CHWs' ability and willingness to work during COVID-19 in Lagos which can help inform how to better support them in their role, strengthening Nigeria's disease response. Despite facing many challenges during the COVID-19 pandemic, CHWs remained devoted to their role in order to protect their communities. However, there is a need for improvement in their work environment. As a country with frequent disease outbreaks, CHWs form an important part of the frontline. Therefore, quick change to ease their strenuous role is needed, to ensure that they can continue to support communities. This could be done through better financial recognition, adequate PPE and transport provision, and by providing more staff. However, international support will be required to ensure this is financially viable and sustainable in the long term. Therefore, collaboration with non-governmental organisations is needed to fund the strengthening of Nigeria's CHW workforce for future disease outbreaks.

## Supporting information

**S1 Fig. Interview topic guide.**
(PDF)

**S2 Fig. PLOS' questionnaire on inclusivity in global research.**
(PDF)

## Acknowledgments

We thank the community health workers interviewed for their support and participation in this study.

## Author Contributions

**Conceptualization:** Zahra Olateju, Tolulope Olufunlayo, Christine MacArthur, Beck Taylor.

**Data curation:** Zahra Olateju.

**Formal analysis:** Zahra Olateju, Beck Taylor.

**Investigation:** Zahra Olateju.

**Methodology:** Zahra Olateju, Tolulope Olufunlayo, Christine MacArthur, Charlotte Leung, Beck Taylor.

**Project administration:** Zahra Olateju.

**Resources:** Zahra Olateju.

**Software:** Zahra Olateju.

**Supervision:** Tolulope Olufunlayo, Christine MacArthur, Beck Taylor.

**Validation:** Zahra Olateju, Beck Taylor.

**Visualization:** Zahra Olateju.

**Writing – original draft:** Zahra Olateju.

**Writing – review & editing:** Zahra Olateju, Tolulope Olufunlayo, Christine MacArthur, Beck Taylor.

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
