## [Decision Letter · Decision Letter 0]

23 Nov 2021

PONE-D-21-32451Community Health Workers experiences and perceptions of working during the COVID-19 pandemic in Lagos, Nigeria - a qualitative studyPLOS ONE

Dear Dr. Olateju,

Thank you for submitting your manuscript to PLOS ONE. After careful consideration, we feel that it has merit but does not fully meet PLOS ONE’s publication criteria as it currently stands. Therefore, we invite you to submit a revised version of the manuscript that addresses the points raised during the review process. Please submit your revised manuscript by Jan 07 2022 11:59PM. If you will need more time than this to complete your revisions, please reply to this message or contact the journal office at plosone@plos.org. Please include the following items when submitting your revised manuscript:A rebuttal letter that responds to each point raised by the academic editor and reviewer(s). You should upload this letter as a separate file labeled 'Response to Reviewers'.A marked-up copy of your manuscript that highlights changes made to the original version. You should upload this as a separate file labeled 'Revised Manuscript with Track Changes'.An unmarked version of your revised paper without tracked changes. You should upload this as a separate file labeled 'Manuscript'.

We look forward to receiving your revised manuscript.

Kind regards,

Khin Thet Wai, MBBS, MPH, MA

Academic Editor

PLOS ONE

Journal Requirements:

a) Did participants provide their written or verbal informed consent to participate in this study?

3. Please include a complete copy of PLOS’ questionnaire on inclusivity in global research in your revised manuscript. Our policy for research in this area aims to improve transparency in the reporting of research performed outside of researchers’ own country or community. The policy applies to researchers who have travelled to a different country to conduct research, research with Indigenous populations or their lands, and research on cultural artefacts. The questionnaire can also be requested at the journal’s discretion for any other submissions, even if these conditions are not met.  Please find more information on the policy and a link to download a blank copy of the questionnaire here: https://journals.plos.org/plosone/s/best-practices-in-research-reporting. Please upload a completed version of your questionnaire as Supporting Information when you resubmit your manuscript.

Additional Editor Comments:

There are minor grammatical errors throughout the manuscript. English language correction is deemed necessary.

Reviewers' comments:

Reviewer's Responses to Questions

**Comments to the Author**

1. Is the manuscript technically sound, and do the data support the conclusions?

Reviewer #1: Yes

Reviewer #2: Yes

2. Has the statistical analysis been performed appropriately and rigorously? 

Reviewer #1: N/A

Reviewer #2: N/A

3. Have the authors made all data underlying the findings in their manuscript fully available?

Reviewer #1: Yes

Reviewer #2: No

4. Is the manuscript presented in an intelligible fashion and written in standard English?

Reviewer #1: Yes

Reviewer #2: Yes

5. Review Comments to the Author

Reviewer #1: This is a very interesting article that is presented and written well. I have a few comments, which I feel would help clarify certain aspects of the study and further enhance the paper:

Line 83: CHW – What training do they get?

Lines 112-121: If a focus group qual study was undertaken in 2020 (Mayfield-Johnson et al) then why is this study being carried out? It would be good to justify the need for this study in a bit more detail. Were there any gaps in the findings from previous studies that led to the research questions/ aims for this study? What would this study add to the current evidence?

Line 131: What is a generic qualitative study? Do you mean it’s not aligned to any particular epistemological/philosophical school of thought? Why was this study design chosen?

Line 144: Why was purposive and snowball sampling used? Any limitations to this approach?

Line 165: “experiences working during the pandemic……” Typo – should say “experiences of working during…..”

Reviewer #2: Dear Editor,

I would like to thank you for the opportunity to review this article. Below are my queries regarding the manuscript. Authors need to read through their article and make some minor grammar corrections with regards to some of their statements. Thank you.

Abstract

Background: reduce some statements and add the aim of the study as a last sentence to the background.

Introduction

Information on paragraph 1 and 2, lines 66-78 may not be necessary. I suggest that the authors focus on the global role of CHWs, their role in Africa and the impact and challenges they are facing in the face of the pandemic.

Line 88; “who” should be deleted.

Statement for lines 98 and 99 should be paraphrased.

Materials and methods

Data analysis

What type of coding frame did you use? Was it hierarchical or flat coding frame? There is need to state the type of coding frame that was used for the analysis.

In your methodology, you made mention that you used inductive and deductive method of coding which is unclear. It is assumed that you start coding your data from the scratch when you use inductive method of coding and all your codes are generated from your data. Also, if iterative process was used in analyzing your data, from my understanding, it means that you used inductive coding while analyzing data and at the same time refining the topics for further interview? How did you combine both inductive and deductive method of coding in analyzing your data since deductive coding has to do with a predefined set of codes that are used in data analysis?

In your methodology, you stated procedures that you used in carrying out the analysis using the thematic method of analysis. I think you should state that seven stages are involved in the frame work method of analysis and explain what was done at each stage for easy comprehension: Transcription; familiarization with the interview; coding; developing a working analytical framework; applying the analytical framework; charting data into the framework matrix; and interpreting the data.

Reflexive process was used to eliminate bias, inductive method which you stated as part of your coding methodology is also a good method in eliminating bias during data analysis. It will be good to add it as part of the method that was used to eliminate bias in your study.

Ethics Statement

Under the ethics statement, it will be nice to state that verbal consent was obtained from all participants after explaining to them the content of the research before commencement of the interview for each participant.

Results

Table 2

1. Influences on ability to undertake COVID-19 Role

a) Trust

b) CHW Knowledge about COVID-19

Looking closely at table 2 theme 1; could it be possible that subthemes

c) Exhaustion

d) Stigma

e) Public misconceptions about COVID-19

f) Delayed Access to Care

g) Lack of Transportation

could be under a new theme tagged “Challenges faced by CHWs”. I think that those challenges that were mentioned was not what influenced the CHWs to take up the COVID-19 role, rather, these were the hitches they encountered while carrying out their duties.

Table 2, already having the themes and subthemes; needs to be further developed by adding columns for “codes” and the “frequency of codes by participants” to further elaborate on the findings of your analysis and for clarity.

Discussion

Paraphrase the statement; lines 534 and 535 "Considering the inevitability of these challenges in any disease outbreak is

important due to their sudden and unfamiliar nature."

Strengths and Limitations

Lines 599 and 600; The majority of participants were female; however, this reflects CHWs being a female dominant profession. The statement needs to be paraphrased. The participants for this study were selected based on purposive sampling, I don’t think is safe to assume that the majority of CHWs in Lagos State or under the LGAs where the study was conducted are female workers except where such data is available.

Reaching data saturation in a qualitative study is import to ensure that justice is done with regards to a particular topic. As part of the limitation of this study, you stated that data saturation was not reached due to the fact that new information was generated during the last interview that was conducted. As part of your recommendation, it is good to state the need for further studies in this regard to explore all options which the current study was not able to cover since data saturation was not reached.

6. PLOS authors have the option to publish the peer review history of their article (what does this mean?). If published, this will include your full peer review and any attached files.

Reviewer #1: No

Reviewer #2: **Yes: **AYI VANDI KWAGHE

---

## [Author Response · Author response to Decision Letter 0]

28 Jan 2022

All responses to reviewer comments are documented in our response to reviewers document.

---

## [Decision Letter · Decision Letter 1]

23 Feb 2022

Community Health Workers experiences and perceptions of working during the COVID-19 pandemic in Lagos, Nigeria - a qualitative study

PONE-D-21-32451R1

Dear Dr. Olateju,

We’re pleased to inform you that your manuscript has been judged scientifically suitable for publication and will be formally accepted for publication once it meets all outstanding technical requirements.

Kind regards,

Khin Thet Wai, MBBS, MPH, MA

Academic Editor

PLOS ONE

Additional Editor Comments (optional):

Every comment is adequately addressed.

Reviewers' comments:

Reviewer's Responses to Questions

**Comments to the Author**

1. If the authors have adequately addressed your comments raised in a previous round of review and you feel that this manuscript is now acceptable for publication, you may indicate that here to bypass the “Comments to the Author” section, enter your conflict of interest statement in the “Confidential to Editor” section, and submit your "Accept" recommendation.

Reviewer #1: All comments have been addressed

Reviewer #2: All comments have been addressed

2. Is the manuscript technically sound, and do the data support the conclusions?

Reviewer #1: Yes

Reviewer #2: Yes

3. Has the statistical analysis been performed appropriately and rigorously? 

Reviewer #1: N/A

Reviewer #2: Yes

4. Have the authors made all data underlying the findings in their manuscript fully available?

Reviewer #1: Yes

Reviewer #2: Yes

5. Is the manuscript presented in an intelligible fashion and written in standard English?

Reviewer #1: Yes

Reviewer #2: Yes

6. Review Comments to the Author

Reviewer #1: I feel that the authors have adequately addressed the reviewers' comments and this is a well written and interesting article.

Reviewer #2: (No Response)

7. PLOS authors have the option to publish the peer review history of their article (what does this mean?). If published, this will include your full peer review and any attached files.

Reviewer #1: No

Reviewer #2: No

---

## [Editor Report · Acceptance letter]

28 Feb 2022

PONE-D-21-32451R1 

Community Health Workers experiences and perceptions of working during the COVID-19 pandemic in Lagos, Nigeria - a qualitative study 

Dear Dr. Olateju:

I'm pleased to inform you that your manuscript has been deemed suitable for publication in PLOS ONE. Congratulations! Your manuscript is now with our production department. 

Kind regards, 

on behalf of

Dr. Khin Thet Wai 

Academic Editor

PLOS ONE